# Hyperbranched Polymers: Recent Advances in Photodynamic Therapy against Cancer

**DOI:** 10.3390/pharmaceutics15092222

**Published:** 2023-08-28

**Authors:** Jie Chen, Yichuan Zhang

**Affiliations:** State Key Laboratory of Antiviral Drugs, School of Pharmacy, Henan University, Kaifeng 475004, China

**Keywords:** hyperbranched polymer, photodynamic therapy, photosensitizer, combination therapy, tumor targeting

## Abstract

Hyperbranched polymers are a class of three-dimensional dendritic polymers with highly branched architectures. Their unique structural features endow them with promising physical and chemical properties, such as abundant surface functional groups, intramolecular cavities, and low viscosity. Therefore, hyperbranched-polymer-constructed cargo delivery carriers have drawn increasing interest and are being utilized in many biomedical applications. When applied for photodynamic therapy, photosensitizers are encapsulated in or covalently incorporated into hyperbranched polymers to improve their solubility, stability, and targeting efficiency and promote the therapeutic efficacy. This review will focus on the state-of-the-art studies concerning recent progress in hyperbranched-polymer-fabricated phototherapeutic nanomaterials with emphases on the building-block structures, synthetic strategies, and their combination with the codelivered diagnostics and synergistic therapeutics. We expect to bring our demonstration to the field to increase the understanding of the structure–property relationships and promote the further development of advanced photodynamic-therapy nanosystems.

## 1. Introduction

Cancer has been a threat to human beings over centuries, causing millions of deaths annually. The battle against cancer has lasted for many decades, although very few effective treatment methodologies have emerged to completely cure it, especially cancers in the developed stages [1]. Surgery, chemotherapy, and radiotherapy are standard options for patients suffering from cancers and can prolong their survival time. However, the limited therapeutic efficacy and serious side effects of such available strategies cannot meet the current medical requirements [2,3]. In this sense, it is essential to develop alternative cancer treatment methods enabling improved therapeutic efficacy and reduced toxicity.

Photodynamic therapy (PDT), which in situ converts light energy into chemical energy, inducing cytotoxicity with the aid of a photosensitizer (PS), has been rapidly developed and utilized clinically [4,5]. Typically, patients who receive PS administration are subsequently illuminated with a focused laser at the desired site of disease. Upon light illumination, non-toxic PSs are excited and transfer the absorbed energy to surrounding triplet oxygens, generating highly cytotoxic reactive oxygen species (ROS) (e.g., singlet oxygen (^1^O_2_) (type I mechanism), superoxide anion (O_2_^•−^), and hydroxyl radicals (^•^OH) (type II mechanism)), causing damage to proteins, nucleic acids, lipids, plasma membranes, and organelles, and inducing cell apoptosis and necrosis and tissue destruction (Figure 1) [6]. A recently developed Nile Blue-derived PS system allows direct energy transfer from excited PSs to RNAs without the generation of any ROS intermediates, enabling an even more efficient PDT treatment, especially in hypoxic tumor regions (type III mechanism) [7]. Because of the non-invasive and spatial- and temporal-controllable features, PDT has been extensively utilized for cancer treatments, introducing direct cancer-cell death, damaging tumorous vasculatures, and activating the immune system, although the application is not limited to oncology [8]. Numerous studies have reported the successful application of PDT in dermatology [9,10], ophthalmology [11], urology [12], cardiology [13], pneumology [14], and dentistry [15]. 

Typically, PSs utilized for PDT are anticipated to be non-toxic and safe for patients. Cytotoxicities are only induced in the region where focused light at the desired wavelength is illuminated, allowing for treatment in a controlled manner. However, most clinically used PSs have intrinsic limitations, such as hydrophobicity, low photostability, an insufficient tissue-penetration depth, and, more importantly, low tumor specificity, restricting their efficacy and causing side effects [16]. In addition, because of the limited diffusion distances of ROS (typically <55 nm) after generation, it is ideal to have the ROS generated intracellularly or on the cell membrane to induce efficient cancer cell damage [17,18]. Therefore, numerous nanomaterials have been developed for the targeted delivery of PSs to tumor tissues and simultaneously improve the aqueous solubility, biocompatibility, blood circulation time, and stability of the loaded PSs [19,20,21,22,23,24]. These nanocarriers can be either inorganic or organic, including carbon [25], silica [26,27], gold [28], metal oxide nanodots [29], polymeric micelles [30], nanospheres [31], liposomes [19], nanogels [32], dendrimers [33], and hyperbranched (HB) polymers [34]. Many of these nanomaterials have been well developed, and their applications in PDT have been extensively reviewed elsewhere [29,35,36]. However, HB-polymer-based nanomaterials possessing many distinct advantages are less discussed compared to other nanosystems [37,38,39,40]. 

In this review, we will focus on the structural properties, synthetic strategies, and applications of HB-polymer-based nanomaterials utilized for PDT against cancer. Synthetic procedures of classic HB polymers used for PDT will be demonstrated first, and the most utilized stimulus-responsive HB polymers for PS delivery will be discussed next. For real medical applications, the surface properties of nanocarriers have to be properly tuned, and thus the surface modification strategies for HB-polymer-based nanoparticles will be introduced here. In the next section, PSs, the other key element in PDT, will be illustrated. Here, we will focus on the most utilized methodologies for PS loading in HB polymers and describe how improvements can be achieved compared to freely used PSs. PSs encapsulated in HB polymers, PSs covalently conjugated to HB polymers, and HB PSs constructed via the covalent conjugation of PSs will be discussed in detail independently, and their advantages and disadvantages will be compared subsequently. By utilizing these HB-polymer-based nanocarriers, chemo-/photodynamic synergistic therapy can be achieved via the simultaneous delivery of PSs with anticancer drugs to tumor regions. The concept of combination therapy utilizing HB polymers will be demonstrated here. We hope this review can stimulate new ideas and inspire continuous endeavors in this research area.

## 2. Hyperbranched Polymers

HB polymers are highly branched, three-dimensional macromolecules with globular and dendritic architectures endowing them with unique physical and chemical properties. Generally, the molecular sizes of HB polymers range from several nanometers to dozens of nanometers, and HB polymers possess high-density functional groups, few chain entanglements, high free volume, low viscosity, and high solubility. In comparison to dendrimers, another important class of dendric polymers with regularly branched and uniform structures, HB polymers synthesized through one-pot polymerizations have a more affordable production route and can be easily scaled up while satisfying their perfect structures. The unique structural properties and availabilities endow HB polymers with considerable potential as delivery carriers for PDT. The intramolecular cavities enable cargo loading, and the high number of surface functional groups provide the possibility of the targeted delivery of cargos to desired regions (Figure 2) [41,42,43,44,45].

In this section, classic HB polymers and stimulus-responsive HB polymers used as PDT materials are discussed, and their functionalization strategies enabling tumor targeting and improving the blood circulation time and biocompatibility are demonstrated.

### 2.1. Hyperbranched-Polymer Synthesis 

Benefiting from the development of synthetic chemistry and polymer chemistry, various synthetic strategies are now available for the construction of HB polymers. Besides Flory’s classic AB_x_ approach [46], and the A_2_ + B_y_-monomer-combination approach, chain-growth polymerization strategies (e.g., self-condensing polymerizations, ring-open polymerizations) have also been utilized for HB-polymer constructions. Considering the biocompatibility and biodegradability, the most frequently used HB polymers for PDT applications are polyphosphates and polyglycerols, which will be discussed in detail here.

Polyphosphates are a class of polymers sharing similar functionalities (phosphoesters) as DNAs and RNAs, and they are widely used as biomaterials in various applications owing to their biocompatibility and biodegradability. Under physiological conditions, most polyphosphates can be degraded into harmless low-molecular-weight products through enzyme-catalyzed mechanisms [47,48]. Therefore, polyphosphates with HB topological structures have received considerable attention and are utilized as nanocarriers for PS delivery [47,48,49,50,51,52,53]. The branched-backbone architectures of HB polyphosphates are typically synthesized via two approaches. One approach is the classic polycondensation reaction using a bifunctional monomer (A_2_) with a trifunctional monomer (B_3_), resulting in a 3D network structure (Figure 3A) [48]. In this case, gelation has to be avoided to generate the desired HB structure, and therefore it is essential that the reaction conditions, including the monomer concentrations, monomer ratio, and order of addition, are precisely controlled [54]. For example, Shi and coworkers optimized the synthetic procedures and successfully synthesized a series of polyphosphate esters via the phosphoryl chloride–phenol condensation and acrylate–piperazine Michael addition reactions [55,56]. Jiang et al. developed a procedure for the synthesis of amphiphilic HB polyphosphoesters via the esterification polycondensation of 1,6-hexanediol (A_2_), phosphorylchloride (B_3_), and linear methoxy-polyethylene glycol (PEG)-OH [57]. The obtained amphiphilic HB polymers can efficiently load hydrophobic drug molecules through self-assembly and are used for tumor-targeted delivery. In addition, stimulus responsiveness can be introduced to the HB polyphosphates using monomers possessing trigger-cleavable linkages. For example, pH-sensitive HB polyphosphates can be constructed using A_2_ monomers containing acetal linkers [52], ROS-sensitive HB polyphosphoesters were synthesized by introducing a thioketal linker to the A_2_ monomer [58], while glutathione (GSH)-responsive HB polyphosphoesters were constructed using a disulfide-containing A_2_ monomer [59]. Similar to the previously discussed synthetic strategies, these cleavable A_2_ monomers were polymerized with phosphoryl chloride (B_3_) to generate stimulus-responsive HB polymers for precise and targeted cargo delivery.

Another well-established fabrication strategy for HB-polyphosphate synthesis is self-condensing ring-opening polymerization utilizing cyclic phosphate monomers [48]. The facile synthetic procedure without the need for a catalyst enables the straightforward production of polymer products with high purities, meeting the requirements of potential materials for biomedical applications. This polymerization strategy was firstly utilized for HB-polyphosphate fabrication in 2009 by Yan et al., in which primary alcohols were used as an initiator for the polymerization with a five-membered cyclic phosphate used as a monomer (Figure 3A). Furthermore, the engineering of the pendant group of the cyclic phosphate monomers with cleavable bonds can endow HB polyphosphates with stimulus responsiveness. Recently, Yan and coworkers designed an ROS-responsive HB-polymer system using a cyclic phosphate monomer containing a hydroxyl and a thioketal unit, showing good ROS-responsiveness, tumor-targeting, and penetration capabilities [53].

HB polyglycerol, a commercially available HB product with excellent biocompatibility and low toxicity, has emerged as a promising HB-polymeric material for biomedical applications [50,60,61,62,63]. HB polyglycerols are polymers covalently connected through ether linkages that contain numerous linear monohydroxy and terminal dihydroxy units, contributing to high hydrophilicity. The high density ether-containing structure together with the hydrophilicity endow such polymers with a reduced unspecific absorption of biomacromolecules and an increased blood circulation time, enabling in vivo medical applications [62,63]. The abundant terminal hydroxy groups on HB polyglycerols can be further transformed to azides, alkynes, or amines, offering potential modification capabilities [64,65]. Typically, HB polyglycerols with controlled molecular weights and dispersities are synthesized through ring-open multi-branching polymerization utilizing glycidol as a monomer (Figure 3B) [60,66]. Initial attempts were carried out by Dworak and coworkers to synthesize HB polyglycerols using cationic ring-open polymerization, although they resulted in insufficient molecular weights and wide dispersities [67]. Sunder et al. reported the anionic polymerization strategy avoiding intra-cyclization side reactions and generated polyglycerols with moderate molecular weights and narrow dispersities [68]. Wilms et al. further overcame the limitation and produced HB polyglycerols with higher molecular weights up to 24,000 Da using a partially deprotonated low-molecular-weight HB polyglycerol as a macroinitiator [69]. Furthermore, rational design using epoxide derivatives containing degradable linkers can afford biodegradable or stimulus-responsive HB polyglycerols. For instance, Kizhakkedathu et al. synthesized a series of pH-sensitive HB polyglycerols using ketal-containing epoxide monomers and investigated the degradable behavior of the materials in acidic conditions mimicking the tumor environment [70]. Similarly, redox-sensitive HB polyglycerols were generated using disulfide containing epoxide monomers through anionic ring-open multi-branching polymerization [71].

Other building blocks, such as polysaccharides, polypeptides, and polyesters, have been reportedly used for HB-polymer constructions, although the combination with PDT has rarely been reported. The synthesis of these HB polymers is not discussed in this review.

### 2.2. Stimulus-Responsive Hyperbranched Polymers

In many delivery systems, PSs are encapsulated in the hydrophobic cavities inside the nanomaterials due to the hydrophobic nature of their large, conjugated structures. This structural feature of such nano-PSs tends to reduce the PDT efficiency due to the limited energy transfer from excited PSs to surrounding oxygen molecules and the restricted diffusion of the generated ROS from the nanomaterial. Therefore, nanocarriers responsive to tumor microenvironments (pH, enzyme overexpressions, high GSH concentrations, etc.) and/or external physical stimuli (light, magnetic fields, ultrasound, temperature, etc.) have been developed to release PSs in tumor tissues to maximize the therapeutic efficacy. During the past few years, stimulus-responsive HB polymers have attracted considerable attention as smart delivery vehicles because of their structural advantages. Till now, numerous stimulus-responsive HB polymers have been designed and utilized for PDT applications (Table 1).

Compared to normal tissues and the blood fluid (pH ≈ 7.4), tumors exhibit a unique acidic environment with pH values ranging from 6.0 to 7.0 due to their abnormal glycolytic metabolism of glucose to generate lactate [72]. The acidic tumor microenvironment and the endosomal (pH = 5.0–6.5) and lysosomal (pH = 4.5–5.0) conditions have been considered ideal triggers for the selective release of cargos in tumor tissue and/or within tumor cells [73]. HB polymers with multiple ionizable moieties (e.g., amines and carboxylic acids) are widely utilized as pH-responsive nanocarriers, which undergo structural changes (swelling, deswelling, and disassembly) and release cargos upon pH changes. Zhang et al. developed a pH-responsive nanomicelle for pyropheophorbide-a delivery using a tertiary amine containing an amphiphilic dendritic block copolymer [74]. When incubated in an acidic environment (pH = 5.4), a swollen structure of micelles was observed, resulting in a significantly higher release rate of pheophorbide than micelles in neutral pH (7.4). Another strategy for pH-responsive HB-polymer construction is to incorporate acid-labile linkers into polymer structures. Acetal [52], β-thiopropionate [75], hydrazone [42,76], and boronate ester [77] are the known acidic-cleavable linker groups and are widely used for pH-sensitive PS nanocarrier synthesis. Wang et al. reported an acetal-containing HB-polyphosphoester nanocarrier loaded with chlorin e6 (Ce6) for pH-activatable PS delivery (Figure 4) [52]. An accelerated release of Ce6 and thus enhanced PDT efficacy was observed in tumor cells compared with normal cells, attributed to the rapid cleavage of acetal linkages under acidic tumor microenvironments.

**Table 1 pharmaceutics-15-02222-t001:** Examples of recently developed stimulus-responsive HB polymers as PS nanocarriers for PDT and PDT/chemotherapy.

Stimulus	Responsive Moiety	Nanocarriers	PS and/or Anticancer Drug	Cumulative Release in Physiological Environment	Cumulative Release Triggered by Internal/External Stimuli	Applications	Reference
pH	Acetal	HB polyphosphate	Ce6	27% release of PS over 20 h at pH 7.4	50% release of PS over 20 h at pH 5.5	PDT	[52]
	Benzacetal	HB polyglycerol	Temoporfin	-	-	PDT	[78]
	Amine	Cellulose nanofibril grafted with HB polyamines	Indocyanine green and doxorubicin	-	-	PDT/chemotherapy	[79]
Redox	Disulfide	HB polyglycerol	Temoporfin	-	-	PDT	[78]
	Disulfide	HB Ce6	Ce6	-	-	PDT	[80]
ROS	Thioether	Synthetic HB polymer constructed using MTPA and TMPTGE	Ce6 and paclitaxel	35.4% release of paclitaxel without light illumination over 24 h	74.8% release of paclitaxel upon 660 nm laser illumination over 24 h	PDT/chemotherapy	[41]
	Diselenide	HB porphyrin	Porphyrin	-	-	PDT/chemotherapy	[81]
	Thioketal	HB polyphosphate	Ce6 and doxorubicin	8% release of doxorubicin without light illumination over 24 h	50% release of doxorubicin upon 660 nm laser illumination over 24 h	PDT/chemotherapy	[51]
	Thioketal	HB polyphosphate	Ce6 and camptothecin	1.1% release of camptothecin without light illumination	10.9% release of camptothecin upon 660 nm laser illumination for 5 min	PDT/chemotherapy	[53]

Abbreviations: MTPA: 3-(methylthio)propylamine; TMPTGE: trimethylolpropane triglycidyl ether.

GSH, the main intracellular antioxidant in mammalian cells, has been found to have abnormally elevated concentrations in tumor cells and is therefore frequently employed as a trigger for tumor-specific cargo release [82]. Disulfide is the mostly studied linkage for GSH-responsive HB-polymer constructions, which can efficiently react with GSH, generating two thiols [78,80,83,84]. By introducing the disulfide linker into HB-polyglycerol structures, Wiehe et al. successfully synthesized a GSH-responsive nanocarrier for directed porphyrin release intracellularly in GSH-rich tumor cells [78]. In the same work, the disulfide bond was replaced by an acid-labile benzacetal linker to generate a pH-responsive nanodelivery system and it was used as a control. In vitro studies using the A431 cell line demonstrate that the GSH-responsive system had a higher PDT efficiency.

However, in some cases, nanodelivery systems responsive to physiological stimuli may suffer from premature drug leakage and toxicity at unspecific sites. A complementary route towards the enhanced spatiotemporal control of therapeutic release is to use drug delivery vesicles that can be triggered by external physical stimuli (e.g., ultrasound or light). During the PDT process, light illumination triggers the generation of ROS, leading to cancer-cell damage. The highly reactive ROS are also an ideal stimulus for the on-demand release of loaded cargos to achieve the spatiotemporal control of synergistic therapeutics [41,51,53,85]. Lee et al. developed a spatiotemporally controlled dual-sensitive nanogel system to improve the PDT treatment efficiency (Figure 5) [43]. A cytotoxic PS, TIr3, was stably encapsulated in disulfide-containing HB-polyglycerol nanogels, which can efficiently release TIr3 via cascade reactions with GSH (reducing disulfide to thiol) and ROS (oxidizing thiol to sulfonic acid). The nanogel can stably hold TIr3 even in intracellular conditions with high GHS concentrations, benefiting from the reversible thiol–disulfide exchange reactions in the nanostructures. Upon near-infrared-light illumination, neutral thiols were oxidized to negatively charged sulfonic acids, inducing electrostatic repulsion and releasing the encapsulated TIr3, enabling precise and targeted PDT by minimizing the premature leakage of toxic PSs.

Although possessing a prolonged blood circulation time and an enhanced tumor accumulation ability, the tumor diffusion and penetration of nanoparticles are often limited due to the existence of sequential pathophysiological barriers of solid tumors [86]. Nanocarriers responsive to tumor microenvironments enabling the transformation of the size and/or surface functionalities are anticipated to allow for efficient tumor penetration and cell-internalization capabilities to overcome complicated biological barriers [87,88,89]. Focusing on this, Yan et al. reported a light-triggered ROS-responsive size-reducing nanoparticle based on an HB thioketal-containing polyphosphoester for enhanced tumor penetration [53]. Upon near-infrared-light illumination, the nanocomposite underwent a remarkable size reduction from 210 to 40 nm, enabled deep tumor penetration, and promoted cellular uptake, resulting in a satisfactory tumor-inhibition efficiency.

### 2.3. Surface Modification of Hyperbranched Polymers

The physicochemical properties and chemical structures of nanocarriers (e.g., surface charges and surface functionalities) were found to have significant impacts on their biodistribution and in vivo fate [37,90]. For instance, although amine-functionalized nanocarriers are known to efficiently penetrate cell membranes because of their positively charged surfaces in physiological environments, the charges induce cytotoxicities and fast removal behaviors in vivo and greatly limit their real medical applications. Negatively charged nanocarriers, in contrast, tend to have prolonged blood circulation times, but exhibit a relatively low cell uptake due to the unfavorable interactions between the nanomaterial and the negatively charged cell membrane. The surface modification of PDT nanocarriers (e.g., PEGylation, carbohydrate conjugation, acetylation, aptamer and peptide conjugation) are key to precise and targeted cancer therapy, enabling tumor accumulation, deep tumor penetration, and high cell internalization [91,92]. When translating to HB-polymer-based nanosystems, the abundant substitutable reactive moieties allow high-density functionalizations, enhancing their biocompatibility and targeting abilities (Figure 6).

A sufficient blood circulation time and stability are of importance for nanomedicine utilized in clinics, allowing drugs to effectively accumulate into tumor tissues through the vasculature [86,90,91,93]. A standard strategy for promoting the blood circulation time for nanomaterials is PEG functionalization. PEG is non-toxic, non-immunogenic, non-antigenic, and highly hydrophilic and has been approved by the FDA for human use [94]. When dosing traditional nanoparticles in vivo, opsonins will be adsorbed on their surfaces and the mononuclear phagocyte system will recognize the nanoparticles as foreign objects and rapidly clear them from systemic circulation. PEGylation can shield the surface from aggregation, opsonization, and phagocytosis, and endow nanocarriers with enhanced hydrophilicity and shielded surface charges, thereby mitigating mononuclear phagocyte system elimination and prolonging blood circulation [62,90,94,95]. PEG functionalization is also applied for HB-polymer-based nanocarriers to improve their in vivo performance and PDT efficacy [42,44,51,81,89,96,97,98,99]. Typically, PEG can be incorporated via physical adsorption or chemical conjugation. In the former approach, PEG is physically coated on the surfaces of nanocarriers through the van der Waals force, hydrophobic interaction, and/or hydrogen bonds [96]. Although they are easily synthesized, it is challenging to achieve structurally stable nanomaterials using such methods. Hence, the covalent conjugation of PEG to particle surfaces is more frequently utilized. Numerous monofunctionalized methoxy PEGs (mPEGs) have been developed for PEGylation, including mPEG-OH [52,81], mPEG-COOH [97], and mPEG-azide [44]. In the work conducted by Wang et al., PEGs were covalently conjugated to the periphery of a carboxylic acid-rich HB polyphosphoester [52]. The PEG shell endowed the nanocarriers with long blood circulation times and high tumor accumulation in vivo, contributing to an excellent PDT therapeutic efficacy. Similarly, Yang and coworkers demonstrated the PEGylation of an ROS-responsive HB polyphosphate using the same reaction condition, affording a self-assembled nanocarrier with excellent stability in cell culture media for over 120 h [51]. Zheng et al. developed a PEGylated dendritic peptide–pyropheophorbide conjugate using a Cu-catalyzed azide–alkyne cycloaddition (CuAAC), allowing for the self-assembly of the amphiphilic HB polymer into a nanostructure, contributing to a satisfactory PDT efficacy [44]. To further improve the tumor penetration capabilities, stimulus-responsive linkers are used for PEG conjugation to achieve size-changeable and surface-functionality-tunable nanocarriers for deep tumor penetrable PDT PS delivery systems. For instance, Li and coworkers developed a size-shrinkable supramolecular-nanohybrid system using an enzyme-cleavable PEG–dendric polymer conjugate (Figure 7) [89]. After accumulation in tumor tissues, matrix metalloproteinases in the tumor microenvironment catalyzed the hydrolysis of the peptide linkages and removed the PEG corona, resulting in a decreased particle size from 130 nm to 50 nm, promoting deep tumor accumulation and penetration.

Apart from PEGylation, various ligands binding to cancer-cell-overexpressing receptors have been attached to nanocarrier surfaces to achieve active targeting to tumor tissues for enhanced PDT specificities. Monosaccharides and oligosaccharides were found to have high affinities to carbohydrate-binding proteins (e.g., glycotransporters and lectins), which overexpress via cancer cells [100]. Therefore, carbohydrate modification is an effective strategy for both targeted tumor imaging and precise cancer treatment [78]. Because monovalent carbohydrate–protein interactions are relatively weak, multivalent glycoconjugates are an attractive strategy to enhance the binding affinity [101]. HB polymers with abundant functional groups are excellent scaffolds for the construction of multivalent glycoconjugates [101,102]. Wiehe et al. developed an antibacterial nanodrug using an HB polyglycerol as the backbone, with mannose and zinc porphyrin conjugated to the terminus of the branched structures aided with CuAAC click chemistry [78]. Benefiting from the high-density conjugation of mannose, the nanodrug system displayed an efficient bacteria-binding capability and thus satisfactory PDT activity. In a following work, an even higher mannose density was achieved using the same chemistry tool, enabling multivalent targeting to A431 cells [78]. With the aid of GSH-degradable nanostructures, the PDT efficacy against lung cancers has been remarkably promoted in comparison with controls using traditional PDT agents.

Hyaluronic acid (HA), a natural anionic polysaccharide, possesses a high affinity to cancer overexpressed biomarkers (e.g., CD44 or RHAMM) and is widely used as a targeting moiety for drug and PDT agent delivery carrier functionalizations [101,103,104]. In physiological conditions, modification using negatively charged HA not only provides nanocarriers a hydrophilic shell to enhance the colloidal stability, but also shields the positive charges of cargo molecules, contributing to enhanced biocompatibility and reduced cytotoxicity [101,102]. Qin and coworkers developed an HA-modified pH-responsive nanocarrier via the electrostatic adsorption of HA on the positively charged HB-polymer nanoparticle surface and used it for targeted delivery to CD44-overexpressed cancer cells [105]. Similarly, Gu et al. developed a CD44-targeted redox-responsive nanocarrier via the electrostatic adsorption of HA on the surfaces of HB-poly(amido amine)/pDNA nanoassemblies and used it for gene delivery towards tumors [106]. Alternatively, surface modifications can be achieved via covalent conjugation of HA to nanocarriers. Jung et al. reported the conjugation of HA to an HB polymer through amide bonds, enabling the targeted delivery of PSs containing HB macromolecules to CD44-positive cancer cells [80]. Additional in vitro and in vivo studies indicated that the HA modification not only elevated the tumor-specific accumulation, but also stabilized the self-assembled nanoconjugates, allowing for precise PDT treatments.

In recent years, aptamers have been increasingly used as a recognition motif for targeted cancer diagnosis and therapy due to their relatively small size, low immunogenicity, excellent specificity, and high affinity to targets [107,108]. Currently, a variety of synthetic chemistry tools have been developed for aptamer conjugation, including the Michael addition [109,110], disulfide bond formation [111], CuAAC [112], strain-promoted azide–alkyne cycloaddition (SPAAC) [113,114], and the inverse-electron-demand Diels–Alder (IEDDA) reaction [65]. Particularly, SPAAC and IEDDA chemistries are the most preferential choices because of their biorthogonality, simple reaction conditions, and high yields. For instance, Yang and coworkers fabricated a self-assembled photoresponsive drug delivery system using DNA-aptamer-grafted HB polymers (Figure 8) [113]. The hydroxy groups on the HB-polymer terminus were in situ transformed to azides, allowing dibenzocyclooctyne-modified Sgc8 aptamer conjugation. The amphiphilic polymer conjugates were assembled into nanoparticles and utilized for targeted delivery towards PTK7-overexpressing cancer cells. Thurecht et al. utilized a similar synthetic methodology to construct ^89^Zr-labeled, aptamer-grafted HB polymers for targeted PET-CT/MRI imaging [114]. Azide-terminated HB polymers were prepared via reversible addition–fragmentation chain-transfer polymerization, and the dibenzocyclooctyne-functionalized ^89^Zr complex and RNV66 aptamer were conjugated to the azide-bearing HB polymer through SPAAC. In comparison to the HB polymers without aptamer decoration, this system demonstrated an enhanced tumor accumulation and thus a promoted therapeutic efficacy. Bohemann et al. reported a tetrazine-strained alkene-based conjugation strategy enabling the aptamer modification of high-molecular-weight HB polyglycerol [65]. Tetrazine-modified Sgc8 aptamers were reacted with trans-cyclooctene-modified HB polyglycerol, generating the desired product for efficient tumor targeting.

## 3. Photosensitizers Utilized with Hyperbranched Polymers

PSs possessing large π-conjugation domains, such as methylene blue derivatives, porphyrins, phthalocyanines, and chlorins, are good PS candidates, allowing long-wavelength-light (>600 nm) absorbance and efficient energy transfer to generate singlet oxygen for PDT [115,116,117]. However, the clinical application of these PSs in PDT is limited by their low solubility and short blood circulation time, which causes poor accumulation at tumor sites and thus results in an unfavorable PDT efficacy and systemic toxicities [115,116,117,118]. In addition, free PSs have the tendency to aggregate in aqueous environments, leading to a decrease in the singlet-oxygen generation efficiency. HB polymers with good biocompatibility and high loading capacities have emerged as promising nanocarriers, protecting PSs from aggregation and degradation, extending their half-life, and improving their tumor accumulation ability [33,35]. Several physical and chemical conjugation strategies have been employed to construct PS–HB polymer nanohybrids for improved PDT efficacy (Table 2).

### 3.1. Photosensitizers Encapsulated in Hyperbranched Polymers

Typically, PSs with the desired structural properties can be entrapped in the cavities of HB polymers or co-assembled with amphiphilic HB polymers to form nanoparticles driven by electrostatic interactions, hydrophobic interactions, and/or hydrogen bonds. In a study reported by Yan et al., an ROS-responsive amphiphilic HB polymer was synthesized using an amine-containing monomer and a tribranched glycidyl monomer through an amine–epoxy coupling reaction (Figure 9) [41]. The resulting amphiphilic HB polymers were subsequently co-assembled with hydrophobic Ce6 and paclitaxel to generate nanomicelles with high loading efficiencies of 53.7% and 26.4%, respectively. Upon light illumination, the nanomicelles uptaken by cancer cells were efficiently disassembled to release Ce6 and paclitaxel to achieve a synergistic treatment. Wang and coworkers reported the Ce6 encapsulation by a PEGylated HB polyphosphate with a loading content of 3.51% [52]. Due to the PEG decoration, the blood circulation time in vivo was efficiently prolonged, with the PDT efficacy promoted compared to free Ce6. Yang et al. employed an ROS-responsive PEGylated HB polymer to simultaneously encapsulate Ce6 and anticancer drug doxorubicin, with loading contents of 3.39% and 3.13%, respectively [51]. Further in vitro and in vivo studies indicated that the resulting nanohybrids realized an amplified synergistic efficacy against drug-resistant tumors.

As discussed in previous sections, HB polyglycerols possess good biocompatibility and long blood circulation times, and they are good candidates as PS carriers for PDT applications. However, because of the highly hydrophilic nature, polyglycerols cannot be directly used for hydrophobic PS encapsulations due to the weak intermolecular interactions. Therefore, hydrophobic segments have been introduced to HB polyglycerols to increase the PS–polymer interactions and promote the loading capacity. From this perspective, Lee et al. developed a GSH/ROS dual-responsive HB-polyglycerol nanogel modified using different hydrophobic cores (benzene, naphthalene, and pyrene) (Figure 5 and Figure 10) [43]. The hydrophobicity was found to have a considerable impact on the PS-loading capacity and the nanogel stability. It was found that the PS-loading efficiency was increased from 11.4% to 29.1% when the benzene hydrophobic core was replaced by pyrene. The enhanced stability demonstrated a dramatically reduced dark toxicity and higher phototoxicity compared to the free PS both in vitro and in vivo.

Electrostatic interactions have been widely applied in the fabrication of nanoassemblies utilizing charged components [125,126]. Many first-line PDT PSs (e.g., Rose Bengal, Ce6, and protoporphyrin) are negatively charged with high singlet-oxygen generation efficiency [118,119,127]. Based on the negatively charged nature of protoporphyrin, Yang and coworkers developed a PS nanodelivery system using a cationic HB polysaccharide as the carrier, and they utilized it for targeted delivery to pancreatic cancer cells [119]. Besides the electrostatic interactions, detailed spectroscopy analyses indicated that the nanoparticles were simultaneously stabilized via hydrophobic interactions and hydrogen bonds. The further electrostatic functionalization of the nanoparticles using negatively charged folic acid enabled the active targeting to cancerous cells to achieve the precise PDT purpose.

Although physical encapsulation provides a simple manufacturing strategy to form PS–HB polymer nanohybrids, this strategy meets some challenges. One disadvantage of this strategy is the limited stability of the nanohybrids in vivo, leading to the premature leakage of encapsulated PSs during blood circulation and the burst release of PSs, causing severe side effects and insufficient therapy efficacy. In addition, it is difficult for physical encapsulation to completely prohibit the aggregation of PSs, leading to a low singlet-oxygen generation efficiency due to aggregation-caused quenching.

### 3.2. Photosensitizers Covalently Conjugated to Hyperbranched Polymers

To overcome the unspecific leakage of PSs and burst-release behavior, PSs can be covalently conjugated to HB polymers to achieve efficient delivery to tumor tissues. Because of the presence of a large number of reactive moieties on HB-polymer peripheries, a classic functionalization strategy is to covalently conjugate PS molecules to the nanomaterial surfaces. In the work reported by Yan et al., Ce6 was chemically conjugated to ROS-responsive HB polyphosphoesters through an esterification reaction, allowing for an extended plasma circulation time, tumor-specific accumulation, and sequentially enhanced PDT efficacy in comparison to free Ce6 [53]. Xiang et al. fabricated a water-soluble HB polyglycerol–PS conjugate via the covalent linkage of carboxylic acid containing PSs (fluorophenylporphyrins) to a hydroxyl-rich HB polyglycerol through a one-step esterification reaction (Figure 11) [34]. Detailed structure–performance relationship evaluations indicated that the HB-polymeric structure possesses remarkable performance advantages in terms of higher biocompatibility and an increased ROS generation efficiency, and promoted phototoxicity in comparison to the linear polymer–PS conjugate or free PS. The reason for the improvement was attributed to the reduced π–π stacking between the PS molecules and thus decreased aggregation-induced quenching. Luo et al. reported a PS (pyropheophorbide-a) conjugation strategy using a thiol–maleimide Michael addition for the functionalization of HB HA [128]. Compared to linear HA, the branched polymeric architecture contributed to a more stable nanostructure, a longer half-life, and a higher tumor accumulation capability, contributing to an enhanced PDT performance in vivo. The same research group also reported a pyropheophorbide-a conjugation strategy using an ester formation reaction for HB PEG functionalization [98]. The amphiphilic PS-PEG conjugate self-assembled into stable nanoparticles in aqueous media with hydrophobic pyropheophorbide-a located at the core and hydrophilic PEG chains dispersed on the periphery. This work demonstrated that by properly tuning the hydrophile–lipophile balances, the stability and physicochemical properties of the self-assemblies can be optimized, contributing to enhanced in vivo therapeutic performances, including blood circulation times, tumor accumulation, tumor penetration, and PDT efficacy. In addition, Luo et al. have shown that PS conjugation can be achieved via amidation reactions by applying carboxylic acid-containing pyropheophorbide-a and primary-amine-containing HB polypeptides [44]. The conjugates were further functionalized via PEGylation through CuAAC chemistry to afford stable and biocompatible supramolecular nanoassemblies with sufficient blood circulation and tumor retention times. The conjugates were found to possess greater resistance to photobleaching while maintaining the singlet-oxygen generation efficiency compared to free PSs, contributing to a remarkably enhanced biosafety and PDT efficiency.

Apart from the conjugation on the periphery of HB polymers, dendric polymers with PS-cored architectures have been developed, albeit with relatively low PS loading. Porphyrin/phthalocyanine-cored dendrimers have been well developed and demonstrated the ability to suppress PS-aggregation-induced quenching and shield their dark toxicities [129]. Benefiting from their high photoactivity, prolonged half-life, good biocompatibility, and high tumor accumulation capability, dendrimer-stabilized PSs have been considered as attractive candidates for PDT applications [130,131,132,133]. However, the stepwise synthesis of dendrimers is time-consuming and expensive, limiting their large-scale production and potential therapeutic exploitations. In contrast, HB polymers that possess similar structural and physicochemical properties can be facilely prepared via one-pot polymerizations, enabling large-scale production for clinical applications [134,135]. Kadhim and coworkers fabricated a porphyrin-cored HB polyglycerol through classic ring-open polymerization using hydroxy-functionalized porphyrin as an initiating seed and glycidol as the monomer (Figure 12) [121]. The porphyrin-cored HB polyglycerols can subsequently self-assemble into spherical aggregates in aqueous media, showing significantly reduced dark toxicity and demonstrating satisfactory PDT efficiency against bladder cancer cells.

To date, chemical conjugation has been a common loading method to form PS–HB polymer conjugates to achieve the efficiently targeted delivery of PSs. However, in some cases, the conjugation of PSs decreases their photodynamic activity. For instance, the Sztandera group found that the conjugation of Rose Bengal to phosphorus dendrimers via tyramine linkers caused a significant decrease in singlet-oxygen generation in comparison with free Rose Bengal [120].

### 3.3. Hyperbranched Polymer Constructed with Photosensitizers

To accomplish the efficient single-oxygen generation of PSs, effective intersystem crossing (ISC) from the singlet state to the triplet state needs to be realized. Polymerization-enhanced photosensitization is an efficient strategy to improve the single-oxygen generation efficiency of PSs. Compared to small-molecule PSs, conjugated polymeric PSs composed of donor and acceptor motifs possess higher light-harvesting abilities and lower singlet–triplet energy gaps, enhancing the ISC efficiency and therefore promoting the singlet-oxygen generation efficiency. To date, it has been found that the HB structure of the donor–acceptor combination is more efficient to improve the ISC process compared to the linear donor–acceptor combination [122]. In addition, the bulk PS macromolecules with HB structures possessing a high density of cavities contribute to enhanced oxygen interactions and thus enable improved PDT efficiency. Recently, Qian et al. reported a dendrimer-structured PS comprising triphenylamine as the donor unit for the connection of multiple BODIPY acceptor units [123]. Compared to the traditional linear donor–acceptor combination, the branched architectures endow PSs with a higher single-oxygen generation efficiency and greater fluorescence quantum yield due to promoted ISC processes. Chen et al. developed a series of aggregation-induced-emission (AIE) active polymeric PSs, including a main-chain polymer, a side-chain polymer, and an HB polymer with the donor and acceptor motifs linked alternatively (Figure 13) [122]. The restricted intramolecular motions and prohibited energy dissipation of the polymeric PSs in their aggregation states contribute to enhanced fluorescence intensities and ROS production yields in comparison to traditional PSs. Among the three types of polymeric PSs, the HB analog exhibited the highest photosensitization efficiency due to its cavity-rich structural features improving the oxygen–PS contact and thus promoting the ROS production efficiency. Further PEG functionalizations of the HB PS enabled tumor-cell-targeted PDT both in vitro and in vivo. Tang and coworkers developed an HB-polymeric PS-construction strategy via the covalent conjugation of the donor and acceptor moieties to restrict the intramolecular motions of the PSs, suppress the nonradiative decay, reduce the singlet–triplet energy gap, and promote the ISC process [124]. The resulting positively charged heteroaromatic HB polyelectrolytes showed high quantum yields and sufficient ROS generation efficiency, and they have been selected as potential candidates for PDT applications. It is worth noting that polymeric PSs show a relatively slow excretion from the body due to the increased molecular weight, so that phototoxicity in normal tissue should be considered carefully from a long-term perspective.

## 4. Chemo-/Photodynamic Combination Therapy

To overcome the limitations and further improve the efficacy of PDT, synergistic therapies combining PDT with other advanced cancer treatment strategies have emerged, showing exciting performances. Among these co-treatments, the combination of PDT with chemotherapy is promising because of the high potential to eradicate cancer cells [136,137,138,139]. As reported, PDT treatment can affect the cell membrane permeability, and adding it as an adjuvant to chemotherapy may increase the deliverability of cytotoxic drugs [140,141,142]. Many studies have demonstrated that chemo-/photodynamic combination therapy can give excellent antitumor performances with the use of relatively low drug doses, even towards chemo-resistant cancers, and simultaneously overcome the limitations of PDT. To realize the PDT/chemotherapy combination, HB polymers are ideal delivery carrier candidates to achieve the efficient coloading and codelivery of PSs and anticancer agents because of the abundant intramolecular cavities, the tailored property, and plenty of functional end groups.

A physical encapsulation strategy for PDT PS and chemotherapeutic codelivery has been reported by Yan et al. [53]. A Ce6-decorated ROS-responsive amphiphilic HB polymer was synthesized and self-assembled into nanoparticles with the clinically used anticancer drug camptothecin loaded in the nanoparticles. These nanoparticles are highly stable under biological conditions and possess long blood circulation times. Upon laser irradiation, the Ce6 loaded in the nanoparticles generated abundant ROS, causing cell apoptosis, and simultaneously triggered the cleavage of thioketal units, reducing the nanoparticle sizes and releasing camptothecin. Moreover, the in situ-generated ROS can destroy the endolysosomal membranes and aid nanoparticles escaping from lysosomes, contributing to the high bioavailability of camptothecin. By combining chemotherapy with PDT, the nanoparticle system exhibited a significantly improved tumor inhibition efficiency compared with individual treatments.

Considering the stability and controllability of drug delivery systems, it is preferable to bring synthetic chemistry tools to covalently bound small-molecule anticancer drugs and PSs to polymeric carriers. Sun and coworkers reported a “Bottom-up” strategy for the construction of an amphiphilic HB poly(prodrug-*co*-PS) unimolecular micelle for chemo-/photodynamic synergistic therapy (Figure 14) [84]. The HB polymers were synthesized via reversible addition–fragmentation chain-transfer (RAFT) polymerization using a redox-responsive camptothecin-based prodrug monomer, a BODIPY-based PS monomer, and a hydrophilic PEG-derived monomer to yield the HB-polymer amphiphiles. The resulting HB polymers assembled into stable unimolecular micelles in aqueous solution and were efficiently taken up by breast cancer cells, releasing camptothecin under the reductive cancerous environment. Upon light illumination, ROS were efficiently generated, contributing to a better therapeutic efficiency than chemotherapy or PDT utilized alone.

An alternative strategy to achieve the stable and controllable codelivery of PSs and anticancer drugs is to individually synthesize a PS-HB polymer conjugate and a drug–HB polymer conjugate and subsequently co-assemble the two components together. Luo and coworkers synthesized an HB PEG–pyropheophorbide-a conjugate and a HB PEG–doxorubicin conjugate and co-assembled the two amphiphilic polymers for chemo/photodynamic combination therapy (Figure 15) [42]. After administration to the acidic tumor environment, the acid labile linker underwent hydrolysis and released doxorubicin to enable the chemotherapy, and light illumination triggered the in situ ROS generation to activate the PDT treatment. To achieve a better and safer therapeutic effect, the ratio of PSs and drugs in the nanoassemblies can be precisely controlled by facilely adjusting the ratio of the polymer–PS and the polymer–drug conjugates. When applying the optimized ratio, monodispersity and good stability can be achieved, and the nanocomposite demonstrated a satisfactory synergistic effect against a 4T1 tumor with an extremely high tumor-growth-inhibition rate (as high as 99%).

## 5. Conclusions and Outlook

Although it has achieved certain successes in clinics, PDT still has some limitations (e.g., the lack of molecular oxygen (hypoxia) in deep solid tumors, insufficient deep tissue penetration of PDT agents and light). Numerous efforts have been made to overcome the barriers (e.g., the construction of codelivery systems containing oxygen-generating components, the development of deep-tissue-penetrable delivery carriers) utilizing advanced optical equipment (e.g., endoscope-like light source), the synthesis of novel PSs, etc. In the design and development of PSs, delivery carriers are usually the key to achieving strong performances. HB polymers are promising candidates as nanocarriers, although they have not received sufficient attention.

This review summarizes recent advances in the design and development of HB polymers as PS nanocarriers for PDT. HB polymers with good biocompatibility, high loading capacities, and ease of fabrication provide a powerful therapeutic nanoplatform for the loading and delivery of PSs to enhance PDT. Several physical and chemical conjugation strategies have been developed for efficient PS loading to improve the PDT efficiency and promote the biocompatibility of PSs. Various surface modifications of nanoparticles have been employed to achieve efficient tumor accumulation. Stimulus-responsive HB polymers have been constructed to efficiently release PSs after reaching targeted tissue in response to internal and/or external stimuli. In addition, HB polymers with the tailored property and plenty of functional groups provide a unique platform for loading PSs in combination with versatile drugs, resulting in a combination of PDT with chemotherapy, demonstrating a synergistic therapeutic effect against tumors.

As discussed above, HB polymers possessing unique structural properties have considerable potential to realize developed and advanced PDT platforms. The following assumptions may contribute to improved PDT performances, although they have not been reported. For example, utilizing the intramolecular cavities of HB polymers, multiple components, including oxygen-generating agents, may be uniformly loaded to increase the oxygen concentrations in hypoxia tumors; the newly developed type III PSs can be utilized with HB polymers to improve the PDT efficiency in hypoxic tumor regions; the decoration of tumor-penetrating peptides on the abundant surface functional moieties of HB polymers and finely tuning the HB-polymer sizes may enable deep tumor tissue penetration to achieve promoted efficacy; hydrophilic HB-polymer encapsulation or conjugation may enable the use of long-wavelength-absorbing PSs with insufficient solubility and/or stability, enabling the utilization of deep-tissue-penetrating long-wavelength light. In short, efforts should focus on the development of HB polymers with precisely controllable structures and functionalities to achieve precise and targeted PDT. We believe the further development of these materials can bring patients an even better treatment efficacy and reduced side effects.

## Figures and Tables

**Figure 1 pharmaceutics-15-02222-f001:**
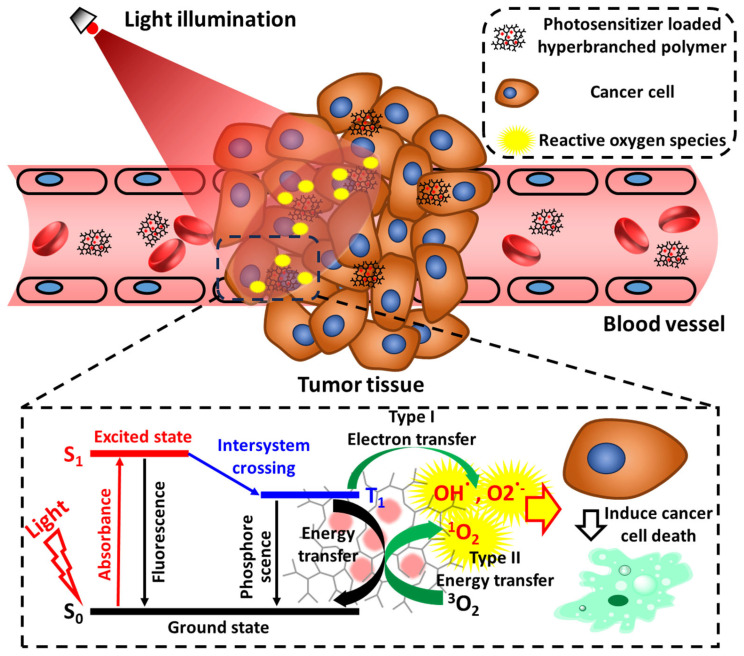
Possible mechanisms of PDT treatment utilizing PS-loaded HB polymers. PSs carried by nanoscale HB polymers are transported to and accumulate in tumor tissue and in situ generate ROS (e.g., ^1^O_2_, O_2_^•−^, and ^•^OH) to induce cancer-cell death with the aid of light illumination.

**Figure 2 pharmaceutics-15-02222-f002:**
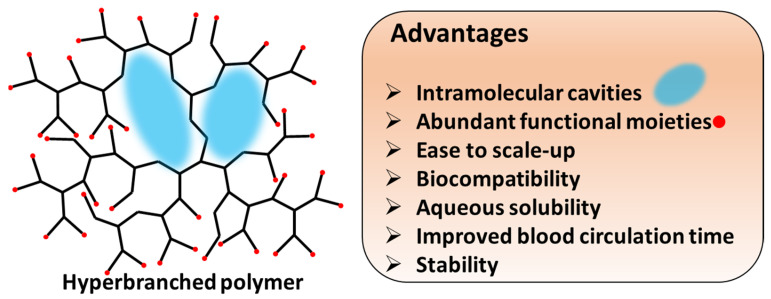
Structural features and advantages of HB polymers as nanocarriers for PDT applications. The potential intramolecular cavities are shown in blue, and the terminal functional moieties are shown in red.

**Figure 3 pharmaceutics-15-02222-f003:**
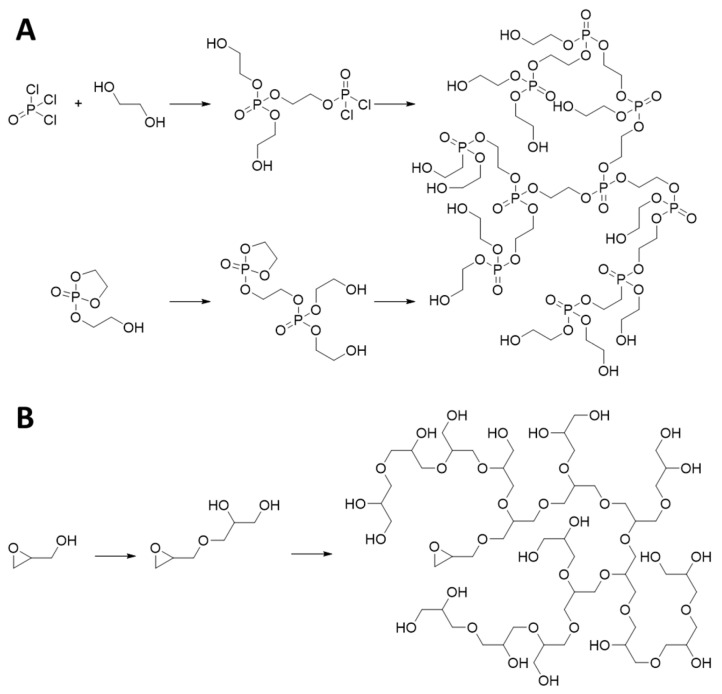
Classic synthetic procedures for HB polyphosphates and polyglycerols. (**A**) HB polyphosphates can be synthesized through a polycondensation reaction using bis-alcohol and phosphoryl trichloride monomers (**upper** procedure). Alternatively, HB polyphosphates can also be synthesized through self-condensing ring-opening polymerization utilizing cyclic phosphate monomers bearing an initiating hydroxy group (**lower** procedure). (**B**) HB polyglycerols can be synthesized through ring-open multi-branching polymerization utilizing glycidol monomers with an initiating hydroxy group.

**Figure 4 pharmaceutics-15-02222-f004:**
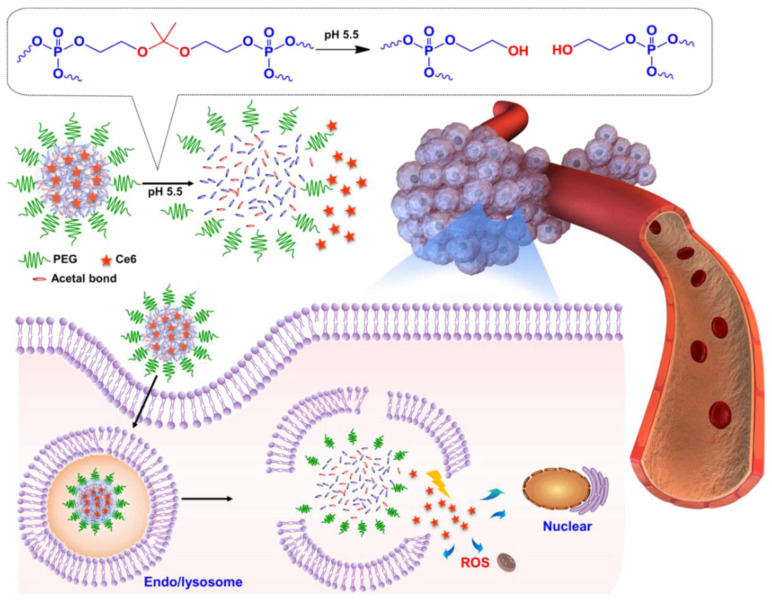
Construction of a pH-sensitive nanodelivery system for Ce6 using an acetal-containing HB polyphosphate. When accumulated in tumor tissues, the acidic microenvironment triggers the cleavage of the acetal linkages in the nanoparticles and Ce6 release, enabling enhanced ROS generation and promoting PDT efficacy. Adapted with permission [52]. Copyright 2018, American Chemical Society.

**Figure 5 pharmaceutics-15-02222-f005:**
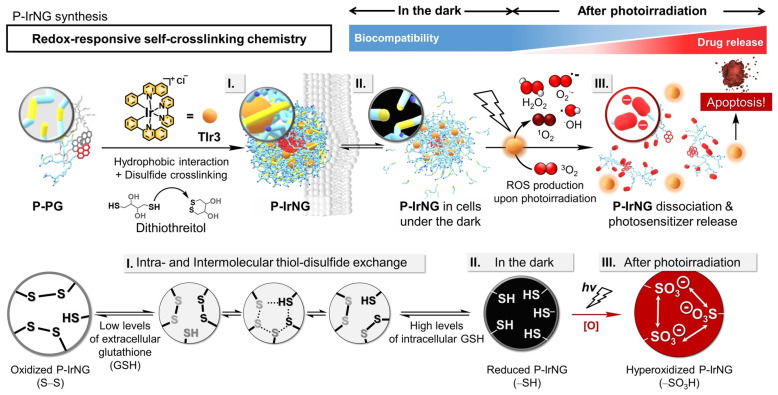
Construction of the GSH/ROS dual-responsive nanodelivery system for cytotoxic PS Tir3 using HB polyglycerol and dithiothreitol. After cell uptake, disulfide bonds are reduced into thiols in the GSH-rich tumor environment. Upon light illumination, TIr3s encapsulated in the nanogels in cancerous cells trigger in situ ROS generation, inducing cell apoptosis, and simultaneously oxidize the thiols to sulfonic acids, destructing the nanogel and releasing the TIr3. Adapted with permission [43]. Copyright 2022, Wiley-VCH.

**Figure 6 pharmaceutics-15-02222-f006:**
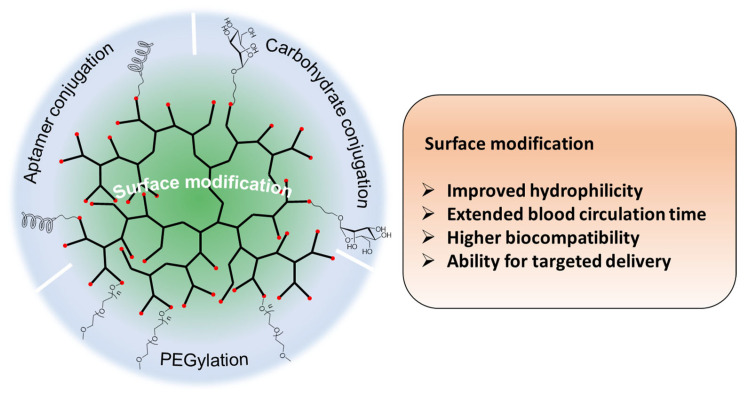
Surface modification strategies for HB polymers and their role in cargo delivery. Proper surface modification of HB polymers can improve their hydrophilicity and biocompatibility, extend the blood circulation time, and offer the ability for targeted cargo delivery.

**Figure 7 pharmaceutics-15-02222-f007:**
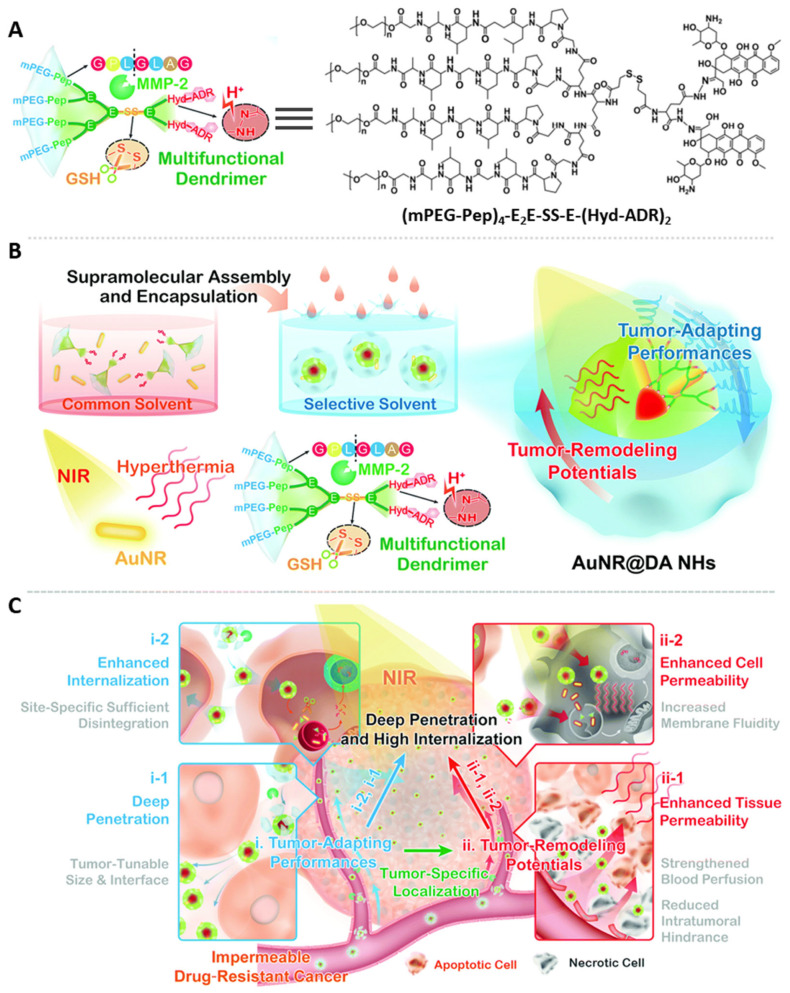
(**A**) Chemical structure of PEG–dendric polymer conjugate synthesized via amidation reactions condensing Boc-protected amino acids and functionalized with PEG and the anticancer drug Adriamycin (ADR). (**B**) Synthesis of tumor-adapting/remodeling nanohybrid using an enzyme-cleavable PEGylated dendric polymer conjugate loaded to gold nanorods. Gold nanorods were initially synthesized using a seed-mediated approach and subsequently mixed with PEG–dendric polymer conjugate to assemble into nanohybrids for further applications. (**C**) Mechanism of actions of the nanohybrid. After accumulation in tumor tissues, the nanohybrids shrink and penetrate into the deep tumor to achieve enhanced PDT efficiency. Adapted with permission [89]. Copyright 2018, The Royal Society of Chemistry.

**Figure 8 pharmaceutics-15-02222-f008:**
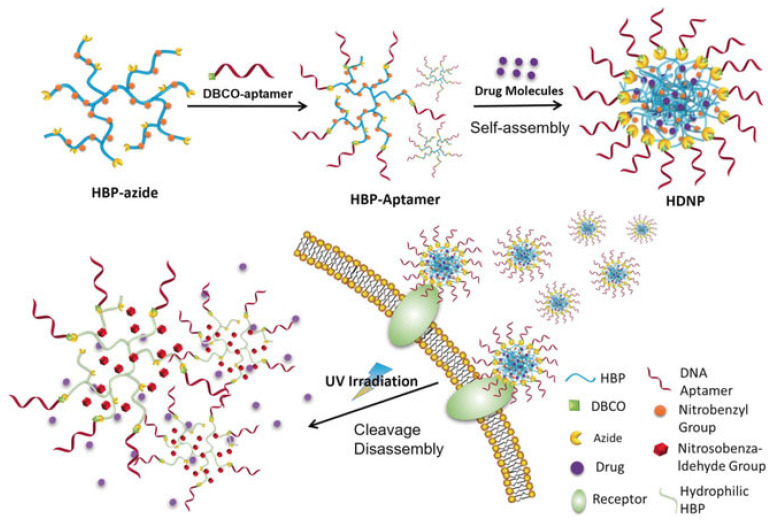
Construction of photoresponsive nanoparticles for targeted delivery of cargo molecules to cancer cells. Dibenzocyclooctyne-functionalized aptamers are grafted to azide-terminated HB polymers through SPAAC. On-demand and controlled release are achieved upon UV irradiation. Adapted with permission [113]. Copyright 2018, Wiley-VCH.

**Figure 9 pharmaceutics-15-02222-f009:**
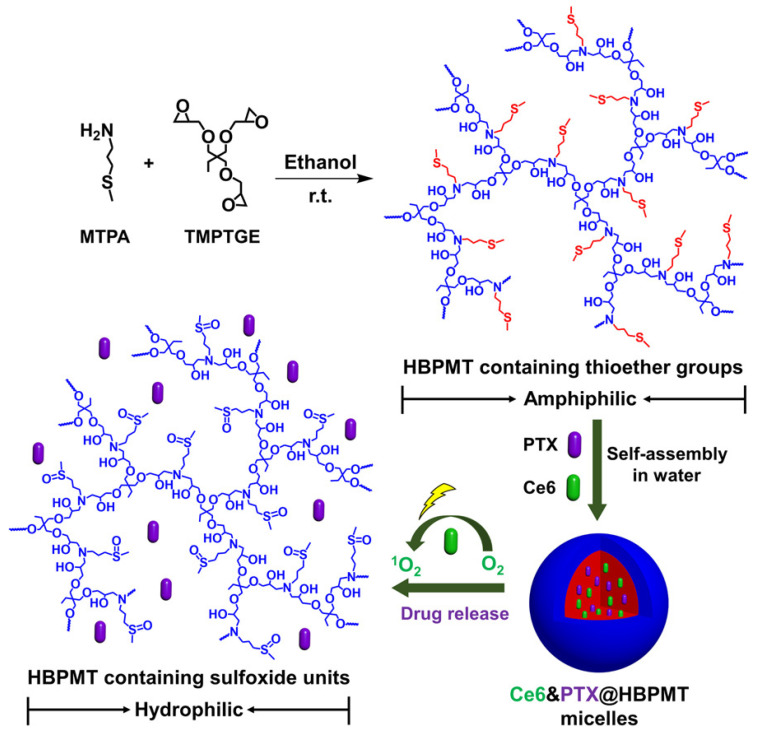
Synthetic route and mechanism of actions of the amphiphilic HB polymers and PS delivery micelles. The HB polymers were synthesized using an amine-containing monomer and a tribranched glycidyl monomer through an amine–epoxy coupling reaction. Upon light illumination, thioethers are in situ oxidized to sulfones, resulting in hydrophobicity changes of the HB polymers and inducing drug release. Adapted with permission [41]. Copyright 2022, Wiley-VCH.

**Figure 10 pharmaceutics-15-02222-f010:**
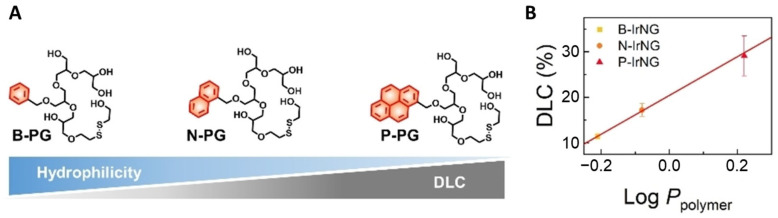
(**A**) Relationships between drug-loading capacity (DLC) and hydrophobicity of HB polyglycerols. (**B**) DLC as function of polymer hydrophobicity. The PS-loading capacity of nanogels gradually increased with the increasing hydrophilicity of HB polyglycerols. Adapted with permission [43]. Copyright 2022, Wiley-VCH.

**Figure 11 pharmaceutics-15-02222-f011:**
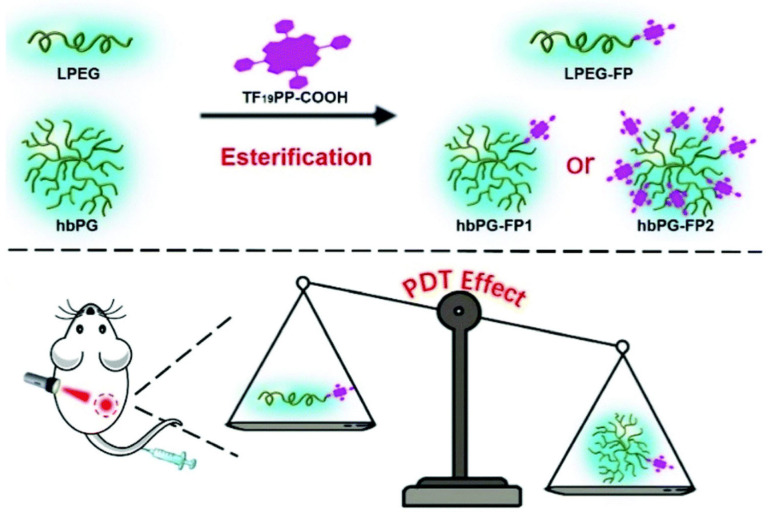
Schematic illustration of the preparation of linear PEG–PS conjugate and HB polyglycerol–PS conjugate. Carboxylic acid containing PS fluorophenylporphyrin was coupled to the hydroxyl-rich HB polyglycerol or monohydroxy PEG through esterification reactions. In vivo studies have shown that the PDT efficiency of HB-polymer-conjugated PSs was greater than the linear polymer analogs. Adapted with permission [34]. Copyright 2020, The Royal Society of Chemistry.

**Figure 12 pharmaceutics-15-02222-f012:**
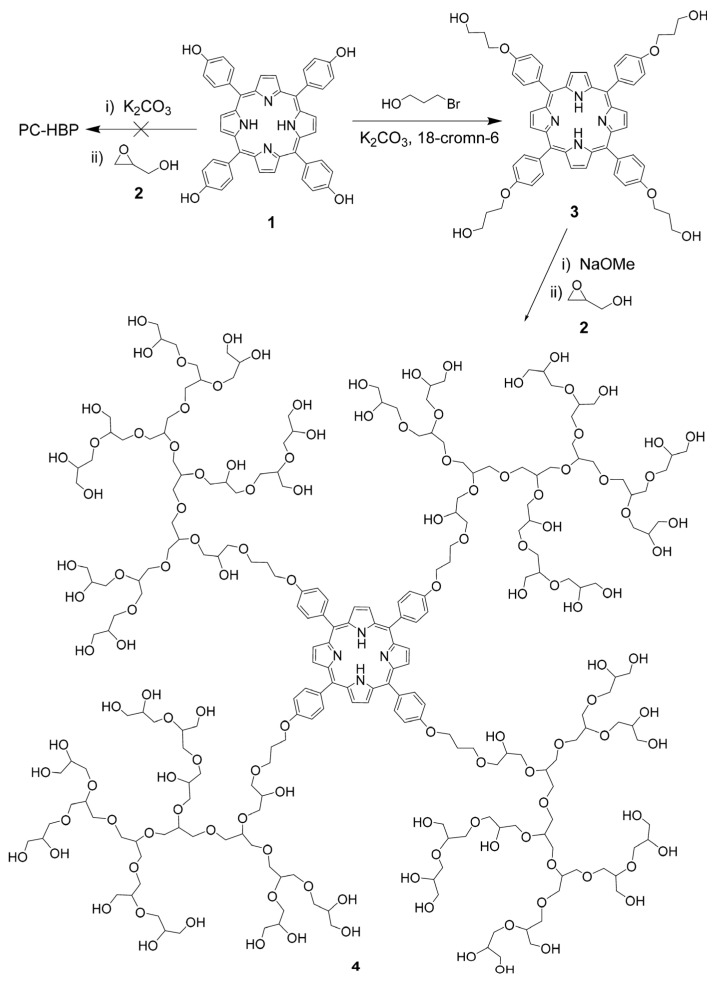
Synthesis of porphyrin-cored HB polyglycerol. The phenol moieties of the porphyrin core were initially extended and converted to aliphatic alcohols and used as an initiating seed for the ring-open polymerization using glycidol as the monomer. Adapted with permission [121]. Copyright 2019, American Chemical Society.

**Figure 13 pharmaceutics-15-02222-f013:**
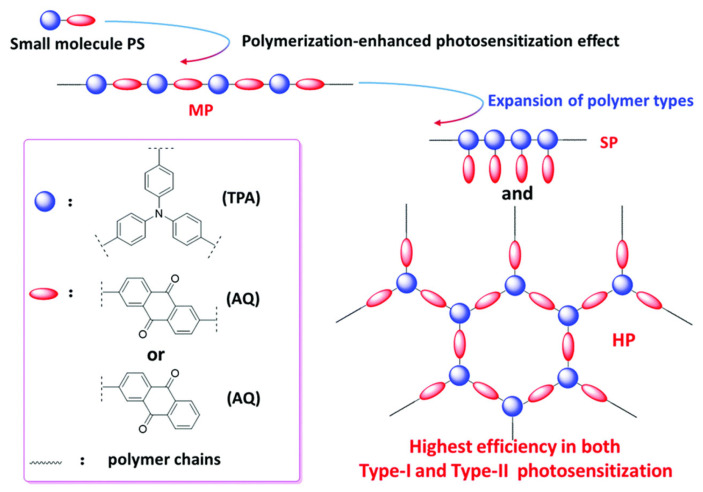
Synthetic procedure of conjugated polymers using PSs as building blocks. Suzuki polymerization was utilized to construct the conjugated polymer. Donor and acceptors are shown in blue and red, respectively. Adapted with permission [122]. Copyright 2022, The Royal of Society Chemistry.

**Figure 14 pharmaceutics-15-02222-f014:**
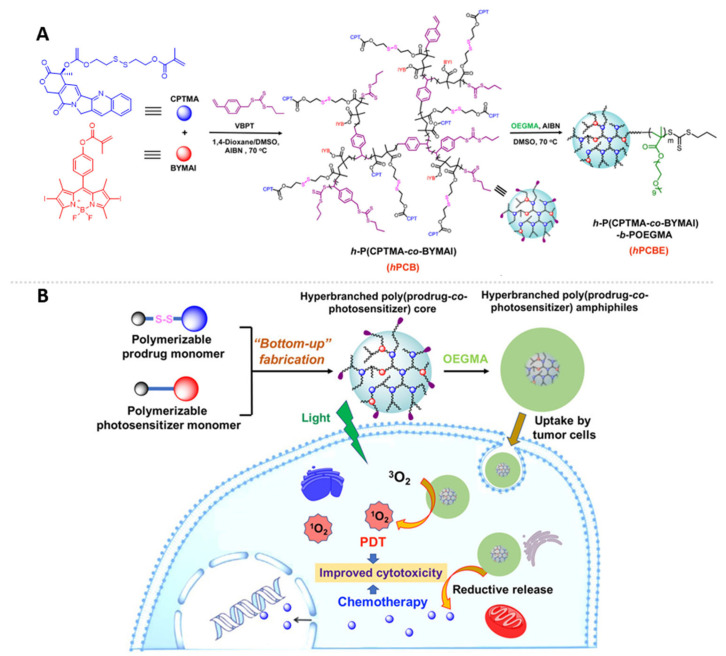
(**A**) “Bottom-Up” construction of hyperbranched poly(prodrug-*co*-photosensitizer) amphiphiles via RAFT polymerization using an anticancer-drug-substituted monomer, a PS-derived monomer, and a PEGylated methacrylate monomer. (**B**) Synergistic effect was achieved via the in situ-generated ROS and the intracellularly released anticancer drugs. Adapted with permission [84]. Copyright 2017, American Chemical Society.

**Figure 15 pharmaceutics-15-02222-f015:**
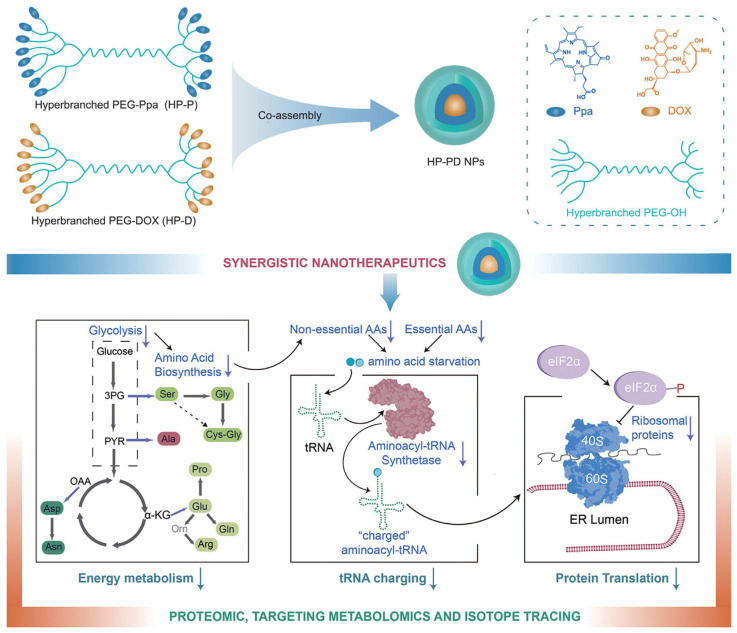
A chemo-/photodynamic combination therapeutic system was established using an HB PEG–PS conjugate and an HB PEG–drug conjugate. The co-assembling of the two components at the desired ratio resulted in a uniform nanotherapeutic agent enabling precise PDT and the controlled release of anticancer drugs. Adapted with permission [42]. Copyright 2022, Wiley-VCH.

**Table 2 pharmaceutics-15-02222-t002:** Comparison of different types of PS–HB polymer hybrids.

Type of PS–HB Polymer Hybrid	Advantages	Disadvantages	Reference
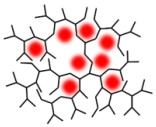 PS encapsulated in HB polymers	High PS-loading capacity Easy fabrication	Unspecific leakage of PSs in blood circulation Burst-release behavior PS aggregation leading to low singlet-oxygen generation efficiency	[41,43,51,52,119]
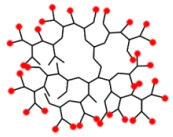 PS surface-conjugated to HB polymers	Chemical and physiological stability Controlled PS-loading content High PS-loading efficiency Mitigated PS premature leakage in blood circulation Potential for on-demand PS release at targeted site	Complicated synthetic procedure and limited candidates of functionalized PS and HB polymers Some covalent linkers decrease photodynamic activity of PSs	[34,44,53,98,120]
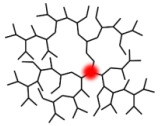 PS-cored HB polymers	Controlled PS-loading content and distribution Negligible PS aggregation Mitigated PS premature leakage	Complicated synthetic procedure and limited candidates of functionalized PS and HB polymers Low PS-loading efficiency	[121]
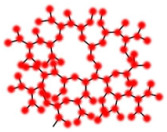 HB polymers constructed by PSs	High ROS generation efficiency Compatible with long-wavelength light source Improved light-harvesting ability	Complicated synthetic procedure and limited candidates of functionalized PS and HB polymers Slow excretion from the body due to increased molecular weight of PSs	[122,123,124]

## Data Availability

Not applicable.

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
