# Peer review of "Hyperbranched Polymers: Recent Advances in Photodynamic Therapy against Cancer"

_pharmaceutics, 2023, doi:10.3390/pharmaceutics15092222_

Round 1

Reviewer 1 Report

Chen and Zhang present in their review article "Hyperbranched polymers" several systems based on polymers with potential application in photodynamic therapy. In general, the English of the manuscript is understandable, although some flaws will have to be corrected for the second round of revisions.

In the introduction, the authors discuss the fundamental aspects of photodynamic therapy. However, I point out several flaws:

1) The authors state in lines 37 and 38 that the energy of the photosensitizer is absorbed by molecular oxygen for the generation of ROS; however, it is necessary to note that this molecular oxygen is in the triplet state. For example, singlet oxygen is also molecular, only in the singlet state.

Furthermore, triplet molecular oxygen is not always necessary for photodynamic activity. New molecules have already been created and are strongly reported in the literature, which, due to type III reactions, do not require this element.

2) The phrase "Benefiting from the advantages, e.g., non-invasiveness, spatial and temporal controllabilities" is not grammatically acceptable. To correct.

3) The hydroxyl radical must have the electron in superscript. This error must be corrected throughout the entire manuscript.

4) The authors state in line 52 that clinically used photosensitizers exhibit cytotoxicity in the dark. Is that true? To my knowledge, they are cytotoxic but at very high concentrations not used in the clinic.

5) Still in the introduction, the authors state that ROS must be produced intracellularly to be efficient. Studies have shown that, for example, if the photosensitizer binds to the plasma membrane, it can cause irreversible damage to it and trigger cell death. You should look for these studies to complete your introduction better.

6) A final paragraph seems missing to present everything that will be discussed next throughout the entire review article.

In the following chapters, the authors present several polymers that can be coupled to photosensitizers, improving their properties such as solubility, targeting, etc. These subchapters were too challenging to read. A review article should contain several tables summarizing the various topics discussed throughout the text. The figures presented are examples of some works and do not represent relevant aspects addressed in the main text. For example, the advantages and disadvantages of each approach carried out in incorporating photosensitizers in the various polymers must be addressed in tables supported with the respective bibliographic references. The text should also be separated into more paragraphs, and irrelevant information should be removed from the manuscript.

In addition, for those outside the area who want to seek more knowledge, the review article must contain complete figure captions. Figure captions should be so full that readers do not need to refer to the text to understand them. An example of this is Figure 3, in which the compounds must all be numbered, and the caption must be more explanatory based on that same numbering. All compounds throughout the manuscript must be numbered and cited in the text or figure captions.

Decidedly, I felt a lack of originality in the manuscript since the authors mostly present figures taken from articles already published (some of which leave something to be desired). I suggest inserting a final, original figure that generally represents the conclusions and future perspectives of the authors at the end of the manuscript.

Finally, the conclusions present little of the author's opinion, except for the last two sentences. In a new paragraph, the authors must add what they really think the future is, how they can do it, and what problems they will eventually have to face.

Provocatively, I question the authors about the novelty that this review brings compared to others recently published with similar subjects: https://doi.org/10.3389/fbioe.2021.783354; https://doi.org/10.1002/adtp.202200165; and why this review urgently needs to be published.

In this way, the authors must review the manuscript in depth, so I accept it with major revisions.

In general, the English of the manuscript is understandable, although some flaws will have to be corrected for the second round of revisions.

Reviewer 2 Report

J. Chen and Y. Zhang reviewed recent applications of hyperbranched polymers as carriers for photosensitizers in photodynamic therapy of cancer. Photosensitizers can be encapsulated or covalently linked to a polymeric core. The topic is hot, and the review is useful.

Issues to be corrected/changed:

Mechanisms of ROS damaging effects to cancer cells should be mentioned in the introductory part.

“Photodynamic therapy” means almost exclusively photodynamic therapy of tumors, with only one example of antibacterial photodynamic therapy, and no examples of antiviral or antifungal therapies. This should be discussed.

Section 2.1 Further functionalization of a dendrimer core should be illustrated with a picture.

PEG is mentioned as a non-immunogenic polymer. Immunogenic properties of other polymers should be discussed.

Figure 14: chemical structures should be added for better comprehension.

Line 298: the structure of the “PEG-dendric polymer conjugate” is not clear from Figure 6. Chemical structures should be added.

Minor issues:

References in square brackets are placed after punctuation marks, commas or periods.

Figure 3A: incorrectly, methylene (-OCH2O-) instead of ethylene (-OCH2CH2O-) is depicted twice.

Reference 28: journal abbreviation should be corrected.

Reference 100: “19” should be superscripted

Reviewer 3 Report

This paper entitled: “Hyperbranched polymers: recent advances in photodynamic 2 therapy” can be accepted after minor revision.

The paper needs more improvement by adding more references.

It is very strange that author did not cite the important following references, it is advised to add the following references: 

1.     B Dhaini et al. Importance of Rose Bengal Loaded with Nanoparticles for Anti-Cancer Photodynamic Therapy, Pharmaceuticals 2022, 15 (9), 1093

2.     B. Dhaini et al., Rose Bengal coupled to AGuIX NPs for anti-cancer photodynamic therapy, 2023, Photodiagnosis and Photodynamic Therapy 41, 103424, ISSN 1572-1000, https://doi.org/10.1016/j.pdpdt.2023.103424

3.     B Dhaini, B Kenzhebayeva, A Ben-Mihoub, M Gries, S Acherar, F Baros, T Hamieh, …, Peptide-conjugated nanoparticles for targeted photodynamic therapy, Nanophotonics, 2021, 10 (2), pp. 3089-3134. https://doi.org/10.1515/nanoph-2021-0275

4.     E. Jamal Al Dine et al. A Facile Approach for the Doxorubicine Delivery in Cancer Cells by Responsive and Fluorescent Core/shell Quantum Dots. Bioconjugate Chemistry 06/2018; 29(7)., DOI:10.1021/acs.bioconjchem.8b00253

5.     Z Youssef et al. Titania And Silica Nanoparticles Coupled To Chlorin e6 For Anti-Cancer Photodynamic Therapy, Photodiagnosis and photodynamic therapy, June 2018 Volume 22, Pages 115–126,

Website: http://www.pdpdt-journal.com/article/S1572-1000(17)30548-3/fulltext   

6.     Z Youssef et al.The application of titanium dioxide, zinc oxide, fullerene, and graphene nanoparticles in photodynamic therapy, Cancer Nanotechnology 8 (6), 1-62, 2017, 2017, 2017, Website: https://link.springer.com/content/pdf/10.1186%2Fs12645-017-0032-2.pdf

7.     E. Jamal Al Dine et al. Synthesis and characterization of smart nanomaterials for cancer treatment. 9th International Conference on Material Sciences (CSM9), Editor T. Hamieh, August 26-28, 2015, ENSIC, Nancy, France, P.170, ENSIC, Nancy, France; 08/2015

8.     Z Youssef et al., Two approaches for elaborating sensitized TiO2 nanoparticles of potential effect in photodynamic therapy, Photodiagnosis and Photodynamic Therapy, 17, A61-A62, 2017. Website: http://www.pdpdt-journal.com/article/S1572-1000(17)30150-3/abstract

9.     He Jia-shuai et al., The Application of and Strategy for Gold Nanoparticles in Cancer Immunotherapy, Frontiers in Pharmacology, 12, 2021, URL=https://www.frontiersin.org/articles/10.3389/fphar.2021.687399, https://doi.org/10.3389/fphar.2021.687399 

On the other hand, the authors are invited to give more information on the physicochemical properties of the Hyperbranched polymers, their surface energy, specific interactions, Florry-Hugghens parameters and their Lewis's acid-base constants.

Round 2

Reviewer 1 Report

I have read and enjoyed the authors’ answers to my questions, suggestions and provocations. The manuscript has been significantly improved. Therefore, I suggest it be published in the Pharmaceutics journal. Congratulations to the authors.

Some minor typos detected.

Reviewer 2 Report

Authors addressed all comments from referees. The only exception is my comment "References in square brackets are placed after punctuation marks, commas or periods." Authors replied: "The reference style has been checked and changed." However, nothing changed - references are still after punctuation marks. Nevertheless, this is purely technical issue, and can be corrected upon editing/proofreading.